# Can The ‘Speed Bump Sign’ Be a Diagnostic Tool for Acute Appendicitis? Evidence-Based Appraisal by Meta-Analysis and GRADE

**DOI:** 10.3390/life12020138

**Published:** 2022-01-18

**Authors:** Ling Wang, Ching-Hsien Ling, Pei-Chun Lai, Yen-Ta Huang

**Affiliations:** 1Department of Surgery, National Cheng Kung University Hospital, College of Medicine, National Cheng Kung University, Tainan 704, Taiwan; jedishellywang@gmail.com; 2Department of Surgery, Hualien Tzu Chi Hospital, Buddhist Tzu Chi Medical Foundation, Hualien 970, Taiwan; lcs8931102@gmail.com; 3Educational Center, National Cheng Kung University Hospital, College of Medicine, National Cheng Kung University, Tainan 704, Taiwan

**Keywords:** speed bump sign, diagnosis, meta-analysis, acute appendicitis

## Abstract

Objectives: The ‘speed bump sign’ is a clinical symptom characterised by aggravated abdominal pain while driving over speed bumps. This study aimed to perform a diagnostic meta-analysis, rate the certainty of evidence (CoE) and analyse the applicability of the speed bump sign in the diagnosis of acute appendicitis. Materials and Methods: Four databanks and websites were systemically searched, and the Quality Assessment of Diagnostic Accuracy Studies 2 was used to evaluate the risk of bias. Meta-analysis was assessed by MIDAS commands in Stata 15. Grading of Recommendations, Assessment, Development and Evaluation methodology was applied to examine the CoE. Results: Four studies with 343 participants were included. The pooled sensitivity and specificity were 0.94 (95% CI (confidence interval) = 0.83–0.98; I^2^ = 79%) and 0.49 (95% CI = 0.33–0.66; I^2^ = 67%), respectively. The area under the summary receiver operating characteristic curve was 0.78 (95% CI = 0.74–0.81). The diagnostic odds ratio was 14.1 (95% CI = 3.6–55.7). The pooled positive and negative likelihood ratios (LR (+) and LR (−)) were 1.84 (95% CI = 1.30–2.61) and 0.13 (95% CI = 0.04–0.41), respectively. According to Fagan’s nomogram plot, when the pretest probabilities were 25%, 50% and 75%, the related posttest probabilities increased to 38%, 65% and 85% calculated through LR (+), respectively, and the posttest probabilities were 4%, 12% and 28% calculated through LR (−), respectively. The overall CoEs were low and very low in sensitivity and specificity, respectively. Conclusion: Current evidence shows that the speed bump sign is a useful ‘rule-out’ test for diagnosing acute appendicitis. With good accessibility, the speed bump sign may be added as a routine part of taking the history of patients with abdominal pain.

## 1. Introduction

Acute appendicitis (AA) is one of the most common causes of acute abdominal pain that requires emergent surgical intervention [1]. However, clinical diagnosis is always challenging, especially in early disease stages. Presenting symptoms such as fever, anorexia, nausea, vomiting and lower abdominal tenderness are usually difficult to distinguish from gastrointestinal or gynaecological diseases. Laboratory tests are usually non-specific parameters for AA diagnosis. Consequently, delayed diagnosis and surgical intervention increases the risk of appendiceal perforation, which can lead to peritonitis or even death. However, negative appendectomy, which is defined as appendectomy revealing a normal appendix upon histological evaluation, may occur in 5–42% of cases [2]. This finding can be associated with considerable morbidity.

The clinical application of different scoring systems, which include symptoms, signs and laboratory examination, have been investigated to enhance the diagnostic value for AA diagnosis [3]. In the latest guideline of the World Society of Emergency Surgery, the experts strongly recommend the use of scoring systems to exclude AA and identify intermediate-risk patients for the need of imaging surveys based on high certainty of evidence [4]. However, some evidence demonstrated that the scoring system was not as reliable as computed tomography (CT) scans [5]. Accordingly, the experts conditionally suggested against the use of Alvarado score for positive confirmation of AA in adults [4]. The routine use of CT scans is not advocated due to high cost, delay in operative intervention and the risk of radiation exposure [6]. In addition, CT scan is not an ideal tool for complicated appendicitis due to low pooled sensitivity published in a meta-analysis [7]. Magnetic Resonance Imaging (MRI) also demonstrated high accuracy in both high pooled sensitivity and specificity, but it costs much more and takes more time for examination [8]. Although the rate of negative appendectomy in the United States has been consistently declining based on the abovementioned strategies, the rate of negative appendectomy remains up to 8.47% [9]. Looking for harmless, costless and accessible indicators to improve the diagnostic test accuracy of AA is still warranted.

Speed bumps, which are frequently used as traffic devices to slow down the speed of vehicles, can be used as a clinical symptom with aggravated abdominal pain while driving over speed bumps (i.e., ‘speed bump sign’) [2]. One of the reasons for pain with AA is due to inflammation of the peritoneum, and it is possible that the impact of going over a bump irritates the parietal layer of peritoneum by stretching or moving as with rebound tenderness [10]. This sign is used by some doctors while obtaining patient history from abdominal pain cases. Pain aggravation while travelling over speed bumps is thought to be associated with an increased likelihood of AA. However, the certainty of its evidence has not been validated. Using systematic review to search and collect all the similar research studies, appraise the risk of bias and further perform meta-analysis if suitable is the standard strategy in the field of evidence-based medicine. The methodological strength of meta-analysis is more efficient to present a combined result by pooling the sample size to increase statistical power than to report the results of each individual study. As such, this study aimed to perform a meta-analysis and analyse the applicability of the speed bump sign through a comprehensive methodology of evidence-based medicine in the diagnosis of appendicitis in patients with acute abdominal pain. 

## 2. Materials and Methods

### 2.1. Search Strategy and Study Eligibility

Our protocol was registered as a systemic review on INPLASY with the following registration number: INPLASY2021110052 (doi:10.37766/inplasy2021.11.0052). The latest statement of the Preferred Reporting Items for Systematic Reviews and Meta-Analysis (PRISMA 2020) [11] and the methodology of the ‘Cochrane Handbook for Systemic Reviews’ [12] were followed. PubMed, Embase, the Cochrane Library and Airiti Library were independently searched by CH Ling and L Wang for studies published before 28 September 2021. The keywords ‘bump*’ OR ‘speed bump’ and ‘appendicitis’ were used, and no language limitation was applied [13]. Relevant articles were also searched in some websites, such as Baidu Schola, ResearchGate, Google Scholar and Google. Duplicated studies and unavailable study data articles were eliminated by human screening. Additional records identified through references were searched. Publications were selected based on describing the following: pain triggered after patients passed speed bumps as a diagnosis of appendicitis; true-positive (TP), false-positive (FP), false-negative (FN) and true-negative (TN) data; and adequate information on derivative parameters (sensitivity, specificity, negative predictive value [NPV], positive predictive value [PPV] and case numbers). After a discussion between CH Ling and L Wang, the final articles for inclusion were determined. 

### 2.2. Methodologic Quality Assessment

‘Quality Assessment of Diagnostic Accuracy Studies 2’ (QUADAS-2) was used to estimate the quality of the enrolled studies [14]. Four aspects of risk of bias (patient selection, index test, reference standard, flow and time) and three fields of applicability concerns (patient selection, index test and reference standard) were appraised following the signalling questions formulated by QUADAS-2. The reading doubts were resolved through a discussion between two reviewers (CH Ling and L Wang). If still inconclusive, the other two authors (PC Lai and YT Huang) were consulted. The figure of QUADAS-2 was constructed by Review Manager version 5.3 (Copenhagen: The Nordic Cochrane Centre, The Cochrane Collaboration, 2014). 

### 2.3. Data Extraction and Statistical Analysis

The following items were independently extracted by CH Ling and L Wang from the enrolled studies: (1) first author’s name, (2) publication year, (3) country, (4) total patient number, (5) gender distribution, (6) patient age, (7) case number and (8) sensitivity/specificity or PPV/NPV. The conversion of a TP/TN/FP/FN case from sensitivity/specificity or PPV/NPV was calculated using Review Manager version 5.3. Inconsistent data were resolved by consensus-based discussion. The pooled estimates of sensitivity, specificity, diagnostic odds ratio (DOR) and positive and negative likelihood ratios [LR (+) and LR (−)] with the corresponding 95% confidence interval (CI) were calculated using the MIDAS command in Stata 15 (StataCorp LLC., College Station, TX, USA) based on the bivariate model [15]. Cochran Q-statistic was used to assess statistical heterogeneity through I^2^ statistic, and heterogeneity across studies was evaluated by the random-effects model. The area under the curve (AUC) was used to describe the overall accuracy as a potential summary of the summary receiver operating characteristic (sROC) curve. Fangan’s nomogram plot analysis for posttest probability was based on the pretest probability and LR (+)/LR (−). Deeks’ funnel plot asymmetry test was conducted to assess publication bias, and *p* < 0.05 was considered statistically significant. Forest plots of pooled sensitivity and specificity, sROC curve, Deeks’ funnel plot and Fangan’s nomogram plot were drawn using the MIDAS command in Stata 15. 

### 2.4. Grading of the Certainty of Evidence

The Grading of Recommendations Assessment, Development and Evaluation (GRADE) methodology was applied to assess the certainty of evidence (CoE) [16]. The CoE can be rated down by one or two levels under the consideration if there are serious or very serious concerns, respectively, in any of the five domains: risk of bias, inconsistency, indirectness, imprecision or publication bias. Sensitivity and specificity were evaluated on the basis of the abovementioned five downgrading domains, and the level of evidence was classified as high, moderate, low and very low. All the authors jointly rated the CoE. GRADE was determined by GRADEpro Guideline Development Tool (McMaster University, 2015 (developed by Evidence Prime, Inc., Hamilton, ON, Canada), accessible from gradepro.org).

## 3. Results

### 3.1. Characteristics of the Enrolled Studies

The PRISMA diagram flowchart of the enrolled articles is illustrated in Figure 1. Seventeen articles (PubMed, 7; Embase, 8; Airit, 0; Cochrane CENTRAL, 0; website, 2) met the initial search criteria, and the references in each article were further screened. One study searched from PubMed discussed abdominal pain experienced when patients passed a road bump, but it provided a different name, i.e., ‘cat’s eye symptom’ [17]. A further search was performed using the term ‘cat’s eye symptom’ to expand our research, but no additional report was found. Two studies were searched from websites, and identical results from the same investigating group were observed [18,19]. One of them was enrolled on the basis of a more detailed description in the manuscript [18]. Finally, four articles were included for meta-analysis [2,10,17,18]. The basic study characteristics are presented in Table 1. All these studies were prospective and published from 1996 to 2020. Two of them were from the United Kingdom, one from Iraq and another from United Arab Emirates. A total of 343 participants were enrolled with a median age range of 25–39 years. Golledge et al. [14] enrolled cases range from the minimum of 4-year-old to the maximum of 81-year-old, while other studies only enrolled adolescence and adults. Pathological confirmation of AA as reference standard was mentioned in all studies. The negative findings of appendectomy were ranged between 5.6% and 46.9%. The sensitivity of diagnosing AA based on the speed bump sign ranged from 80% to 97%, and its specificity varied from 30% to 52%. 

### 3.2. Quality of the Enrolled Studies

The methodological quality of the included studies evaluated with QUADAS-2 is shown in Figure 2, where the risks of bias and applicability concerns are also presented. In terms of ‘avoid inappropriate exclusion’ in QUADAS-2, all the studies did not clearly mention the exclusion criteria for patient selection. Cases with possible diagnosis of AA to ask related questions on the speed bump sign in the enrolled studies were defined vaguely. Therefore, an unclear risk of bias was rated in the aspect of patient selection amongst all the four studies. In Haider et al. [18] and Ashdown et al. [2], the included patients were already assessed by the physicians and considered to have AA, so a high risk of bias in the domain of patient selection was evaluated. In two studies [2,18], a questionnaire survey was performed from triage until cases were transferred to a theatre to minimise recall bias. Golledge et al. [17] and Haider et al. [18] did not report the time of having the questionnaire survey, which yielded an unclear risk of bias in the domain of the index test. The histological diagnosis of appendicitis as the reference standard was defined in all included studies, which was not considered to elicit risk of bias. Moreover, no risk of bias was found in the domain of flow and timing. Regarding applicability, low concerns were judged in the domain of patient selection, index test and reference standard.

### 3.3. Pooled Estimates of Sensitivity/Specificity, sROC and DOR

For the accuracy in predicting appendicitis amongst patients who had increasing abdominal pain while travelling over speed bumps, the DOR was 14.1 (95% CI *=* 3.6–55.7). Pooled sensitivity and specificity were 0.94 (95% CI *=* 0.83–0.98; I^2^
*=* 79%) and 0.49 (95% CI *=* 0.33–0.66; I^2^
*=* 67%), respectively (Figure 3). Linear regression for sROC was generated after the mathematical manipulation of true and false positivity (1-specificity) of each study (Figure 4) [20], and the AUC of sROC was 0.78 (95% CI *=* 0.74–0.81). 

### 3.4. CoE by GRADE Methodology

The first domain of GRADE was downgraded by one level because of some risks of bias evaluated in all included studies. The CoE of sensitivity and specificity was downgraded by one level because of high heterogeneity (I^2^ > 50%). Similarly, downgrading by one level was due to a wide range of 95% CI in pooled specificity. Deeks’ funnel plot asymmetry test showed nonsignificant findings (*p*
*=* 0.89, Figure 5), which indicated no publication bias. The overall certainties of evidence were low and very low in sensitivity and specificity, respectively (Table 2). The number of TP/FP/TN/FN per 100,000 patients tested on the basis of the latest worldwide epidemiology of AA 21] is also listed in Table 2. 

### 3.5. Fagan’s Nomogram Plot Analysis

In this study, pooled LR (+) and LR (−) were 2.00 (95% CI *=* 1.30–2.61) and 0.13 (95% CI = 0.04–0.41), respectively. Fagan’s nomogram plot, a graphic tool, was used to estimate the change in the probability that our patients had AA. A line was drawn starting from pretest probability and connected to LR (+) and LR (−). Thereafter, the line was extended to the right until the posttest probability was reached. The intersection point was set as the new estimated probability, which showed that the patient had a curtained outcome or disease. In our study, when the pretest probabilities were 25%, 50% and 75% based on the physician’s clinical judgement, the posttest probabilities referring to LR (+) were 38%, 65% and 85%, respectively, and the posttest probabilities referring to LR (−) were 4%, 12% and 28%, respectively (Figure 6). In summary, the possibility of AA in a patient without speed bump sign would be less likely whether high or low probability after initial judgement. In contrast, the possibility of AA in a patient with speed bump sign only raised little confidence for definite diagnosis.

## 4. Discussion

In this meta-analysis for diagnostic test accuracy, the speed bump sign provided an easy indicator in predicting AA upon the arrival of a patient with abdominal pain at an emergency room. The DOR ratio is a single indicator of how informative a diagnostic test is that is independent of the prevalence of the disease/disorder [21]. Higher DOR may be indicative of better test performance. In the past meta-analysis, pooled DORs of various indicators for AA diagnosis have been reported, such as Alvarado score (7.99) [22], Raja Isteri Pengiran Anak Saleha Appendicitis (RIPASA) score (24.66) [22], neutrophil-to-lymphocyte ratio (14.34) [23], procalcitonin (21.4) [24], abdominal ultrasound (6.88) [25], CT (129.6) [8] and MRI (129.6) [8]. CT and MRI are without doubts the most accuracy tool for AA diagnosis. The pooled DOR of speed bump sign in our study yielded 14.1, which may be an acceptable value because the information could be obtained just by taking history. However, the disadvantage of DOR is the impossibility to weigh the TP and FP rate separately [21]. In a test with extreme heterogeneity in sensitivity and specificity, the diagnostic value of this test should be inspected separately.

If a test displays a high sensitivity, it will detect the disease or disorder with confidence; if the results of the test are negative, there is certainty that no disease or disorder is present. Therefore, a high sensitivity test helps to rule out the disease/disorder when the result is negative, which is called the mnemonics of SnNout [26]. On the other hand, the mnemonics of SpPin indicates that a high specificity test helps rule-in a disease/disorder with a high degree of confidence if the result is positive. Based on the pooled estimates in our study, the high sensitivity of increasing pain while driving over speed bumps is a basis for yielding a strong rule-out value to exclude AA. The pooled sensitivity of 94% is even better than ultrasound (77.2%, 95% CI = 75.4–78.9%) [25] in a previously published meta-analysis. Since 1980, many score systems have been developed for the diagnosis of appendicitis, and the most widely used system is Alvarado score. This system, including eight parameters with clinical symptoms and laboratory data (migration of pain, anorexia, nausea, tenderness over right lower quadrant, rebounding pain, elevated body temperature, leukocytosis and shift of white blood cell count to the left) is considered a reasonable and simple system that can be used easily in clinics or emergency departments [27]. However, the pooled sensitivity of the Alvarado score for the diagnosis of appendicitis is only 69% (95% CI = 67–71%) in a recent meta-analysis [22]. Another system is Raja Isteri Pengiran Anak Saleha Appendicitis (RIPASA) score, which consists of two demographic information (gender and age), five symptoms (right iliac fossa pain, migration of right lower quadrant pain, anorexia, nausea and vomiting and duration of symptoms), five signs (right iliac fossa tenderness, right iliac fossa guarding, rebound tenderness, Rovsing’s sign and elevated body temperature) and two laboratory data (raised white blood cell count and negative urine analysis); it has been considered the most accurate scoring system for AA diagnosis [28]. The pooled sensitivity of the speed bump sign is similar to RIPASA score (94%, 95% CI = 92–95%) [22], indicating the value of the speed bump sign in clinical applications. In addition, unnecessary CT scans can be avoided because of the similar pooled sensitivity between speed bump sign and CT (95%, 95% CI = 93–96%) [29]; consequently, medical cost and radiation exposure can be reduced. Although a low CoE was evaluated by GRADE methodology in pooled sensitivity because of some RoB and heterogeneity, taking the history of the speed bump sign in cases of suspected AA is strongly recommended. 

The low specificity (49%) of the speed bump sign indicates that patients do not definitely have AA, although they experience aggravating pain when they pass speed bumps during travel. In fact, the specificity of RIPASA score is also low (55%, 95% CI = 51–59%) [28]. Based on the abovementioned mnemonics of SpPin, positive findings from a tool with high specificity may be more suitable to rule-in the disease [26]. Therefore, further examination with high specificity in a case with a positive speed bump sign should be performed. Although the Alvarado score has a better pooled specificity (77%, 95% CI = 74–80%) than other scoring systems [22], its value cannot persuade surgeons to decide on performing an appendectomy. In the 2020 guidelines of the World Society of Emergency Surgery, the experts strongly recommend the use of the ‘Appendicitis Inflammatory Response Score’ (AIRS) and the ‘Adult Appendicitis Score’ (AAS) as clinical predictors of AA [4]. However, systematic reviews of AIRS and AAS have not been reported and the strength of evidence in AIRS and AAS should be re-evaluated. Abnormalities in computed tomography may remain important information for making surgical decisions on patients with suspected AA based on a better pooled specificity (94%, 95% CI = 92–95%) in meta-analysis [29]. 

Considering the incidence of 228/100,000 as the pretest probability [30] in Table 2, only 14 cases present as false-negative amongst 48,902 cases suspected of having AA and not accompanied with a speed bump sign. It indicates again that speed pump sign can be a good tool for AA screening in the triage of emergency medical services. However, applying LR for clinical judgement may be more useful in daily practice [31]. LR represents how much more likely a diagnostic tool is amongst people who have specific clinical presentation than amongst people who do not have the presentation [31]. The pretest probability of an individual case may rely on a physician’s subjective experience and objective information, such as physical examinations, laboratory tests and image findings. Weighted judgement, or posttest probability in statistics, can be changed following the consideration of LR. As shown in Figure 6, there is a slightly increased certainty in AA if a patient feels more pain when he/she goes over a speed bump when the value of AA probability is suspected by a physician. However, it still provides additional confidence for the AA diagnosis, which might be beneficial to utilise the speed bump sign in limited resource areas. By contrast, a large decrease in the probability of AA is depicted in Fagan’s nomogram plot if a patient has no strengthened tenderness. Andersson et al. [32] also investigated the diagnostic merit of different clinical features related to AA through a meta-analysis. In our study, the LR (−) of pain caused by a speed bump was 0.13, which outperformed several parameters, such as migratory pain (0.52), nausea and vomiting (0.72) and rebounding tenderness (0.39) [32]. Indeed, the results in our study demonstrated that the negative finding of the speed bump sign in a patient with abdominal pain can be applied as a strong hint to exclude the diagnosis of AA. However, probabilities of 28% are still not good enough when the pretest probability is as high as 75% with negative speed sign in our study. We believe that combining with the RIPASA score and speed bump sign may provide higher sensitivity for AA diagnosis. Procalcitonin, an indicator for systemic bacterial infection, provided better pooled specificity than sensitivity for AA diagnosis [24]. Therefore, a new scoring system including both speed bump sign and procalcitonin might be more helpful. Such evidence should be confirmed by future research.

The speed bump sign per se and this systemic review had several limitations. Firstly, abdominal pain exacerbated while passing over a speed bump was a subjective feeling. The four included studies gathered information from patients by using a questionnaire, and pain might be overestimated by some patients due to recall bias. To obtain the correct answer from pre-school children is also a problem. Therefore, the accuracy of the speed bump sign for paediatric AA diagnosis should be investigated. In addition, the age of patients in the four included studies was relatively young; therefore, the results in our study might not be suitably applied to the elderly patients who might have had sensory abnormalities. Secondly, the criteria of the enrolled or excluded cases were not declared in detail. Other important abdominal diagnoses, such as a ruptured ovarian cyst, pelvic inflammatory disease and diverticulitis, might be presented as a positive speed bump sign; however, some of these might be considered after obtaining patient history and gender consideration. Prior speculation might interfere the diagnostic value of the speed bump sign. Thirdly, the total number of participants in these four studies was only 343. The number of available studies, cases and experience from countries would limit this study. Due to limited cases, there were no subgroup results between complicated and non-complicated AA in the included studies. Based on the mechanism of peritoneal irritation during the pass of speed bumps, aggravation of abdominal pain might be more strongly triggered in patients with complicated AA. Perhaps the sensitivity might be higher in complicated AA than non-complicated AA. Of course, we are not sure the different presentation of severity in speed bump sign between two groups. To diagnose complicated AA is an annoying issue for physicians, and future studies are warranted to determine the diagnostic value of speed bump sign for patients with complicated AA. Evaluation of speed bump sign takes only a few times, and appendicitis is a common disease in general as well as for paediatric surgery. Repeated studies can be easily and meaningfully carried out. More rigorous and large-scale studies should be conducted to further determine the strength of evidence of the speed bump sign for the diagnosis of AA. Lastly, presenting the CoE in a systematic review with meta-analysis is a widely promoted and encouraged issue in evidence-based medicine. We only depicted the CoE of sensitivity and specificity in layer one, which indicated the levels of evidence in accuracy of a diagnostic test. Recently, the GRADE Working Group suggested identifying three types of layers of evidence summaries [33]. The layer two aims to describe the direct undesirable effects of a test or direct burden from the test, which may not be concerned in the speed bump sign. Layer three includes information for the outcomes following a decision analysis; insufficient data were available in the present studies. The final goal of evidence-based medicine to apply test accuracy to patient-important outcomes and provide recommendations [34] as well as evidence for the decision framework designed by the GRADE Working Group may be used to present the most comprehensive view about the significance of the speed bump sign for AA diagnosis if more relevant studies are published in the future [35].

## 5. Conclusions

In our study, the speed bump sign provided very high sensitivity and very low LR (−), which could be considered as a useful tool to exclude AA if not mentioned by patients. However, further examinations are still needed for making surgical decisions on patients with a positive presentation of the speed bump sign. Uneven road surfaces or potholes on the ground may provide similar effects to those of a speed bump, with the former more often being encountered. Considering it a common phenomenon when a patient heads to a hospital, questioning about the ‘speed bump sign’ should be added to the routine questionnaire when doctors take history from patients with abdominal pain.

## Figures and Tables

**Figure 1 life-12-00138-f001:**
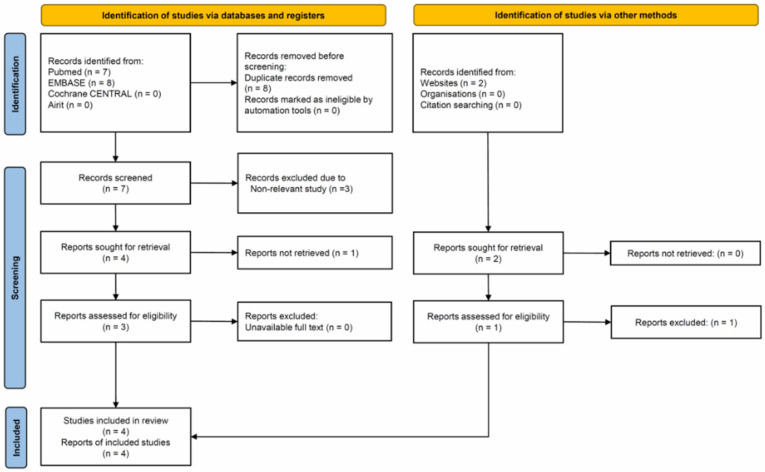
Flowchart of included and excluded studies.

**Figure 2 life-12-00138-f002:**
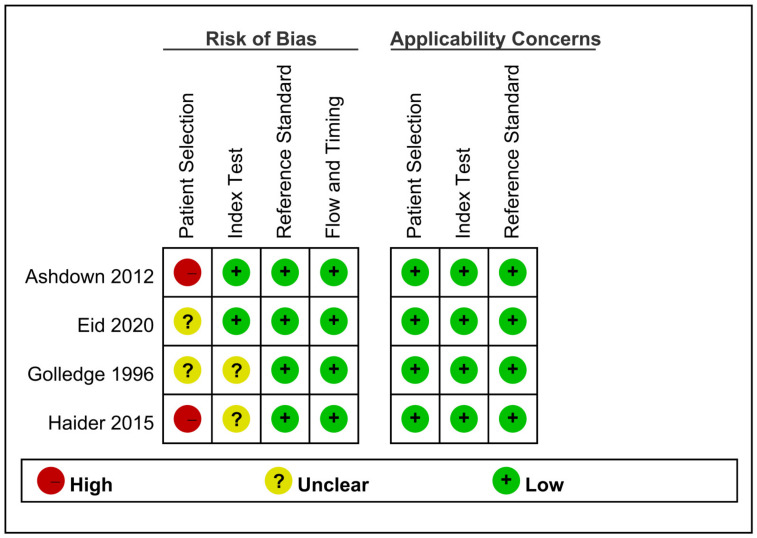
Quality of the enrolled studies appraised by QUADAS-2.

**Figure 3 life-12-00138-f003:**
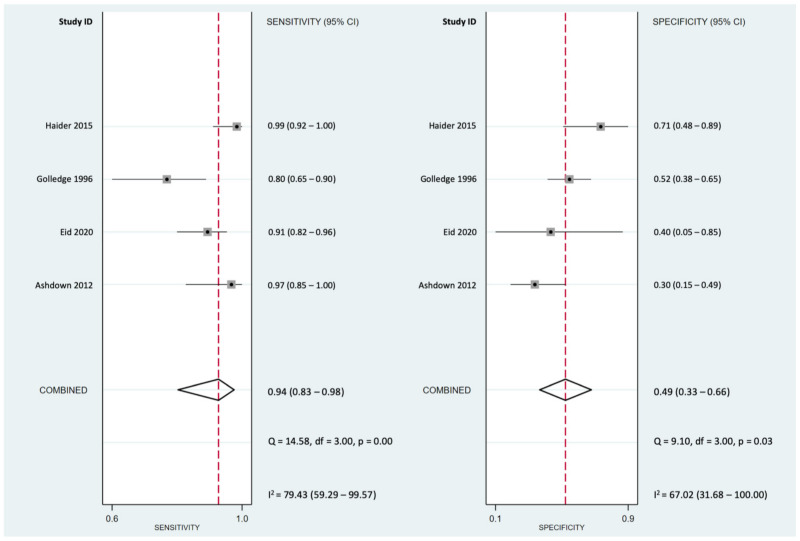
Forest plots of the pooled sensitivity and specificity for a speed bump sign in the diagnosis of acute appendicitis.

**Figure 4 life-12-00138-f004:**
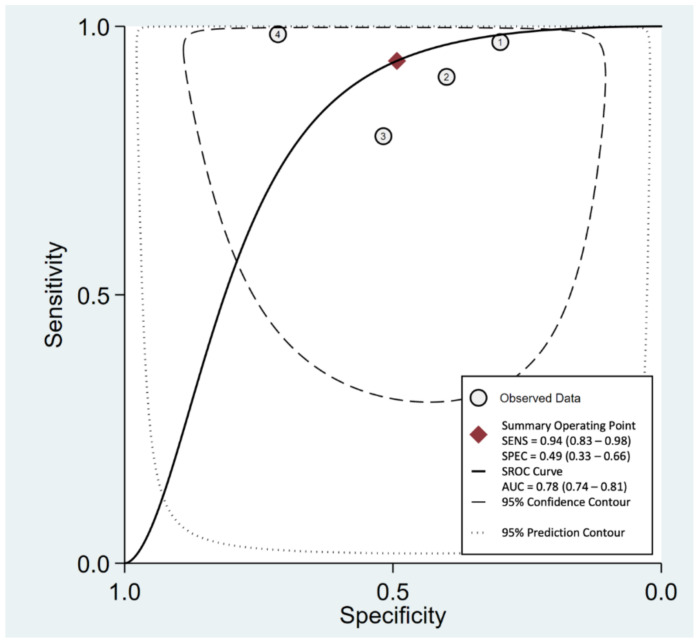
sROC curve of the speed bump sign in the diagnosis of acute appendicitis.

**Figure 5 life-12-00138-f005:**
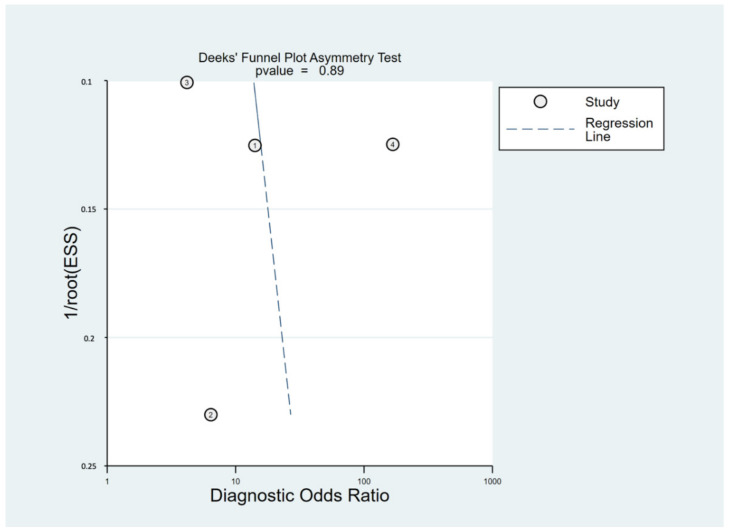
Deeks’ funnel plot asymmetry test of the included studies.

**Figure 6 life-12-00138-f006:**
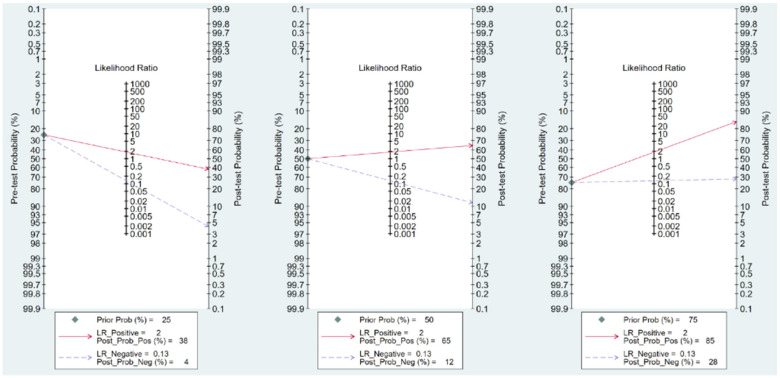
Fagan’s nomogram plot to estimate the change in probability of whether our patients had appendicitis. LR: likelihood ratio, Prob: probability, Pos: positive, Neg: negative.

**Table 1 life-12-00138-t001:** Basic characteristics of included studies.

Study	Golledge et al.	Ashdown et al.	Haider et al.	Eid et al.
Year	1996	2012	2015	2020
Country	United Kingdom	United Kingdom	Iraq	United Arab Emirates
Study design	Prospective	Prospective	Prospective	Prospective
Sample size (Male/female)	100 (39/61)	64 (NA/NA)	89 (NA/NA)	90 (65/23)
Median age (years) Range (years)	25 (4–81)	34 (17–76)	39 (16–65)	34 (15–53)
Sensitivity	0.80	0.97	0.97	0.90
Specificity	0.52	0.30	0.30	0.40

**Table 2 life-12-00138-t002:** Certainty of evidence by GRADE methodology.

Question: Should ‘Speed Bump Sign’ Used to Diagnose Acute Appendicitis in Emergency Department?									
Sensitivity			0.94 (95% CI: 0.83 to 0.98)						
Specificity			0.49 (95% CI: 0.33 to 0.66)						
Prevalence			0.228%						
Outcome	№ of Studies (№ of Patients)	Study Design	Factors That May Decrease Certainty of Evidence					Effect per 100,000 Patients Tested	Test Accuracy CoE
			Risk of Bias	Indirectness	Inconsistency	Imprecision	Publication Bias	Pre-Test Probability of 0.228%	
True positives (patients with acute appendicitis)	4 studies 343 patients	cross-sectional (cohort type accuracy study)	serious ^a^	not serious	serious ^b^	not serious	none	214 (189 to 223)	⨁⨁◯◯ Low
False negatives (patients incorrectly classified as not having acute appendicitis)								14 (5 to 39)	
True negatives (patients without acute appendicitis)	4 studies 343 patients	cross-sectional (cohort type accuracy study)	serious ^a^	not serious	serious ^b^	serious ^c^	none	48,888 (32925 to 65,850)	⨁◯◯◯ Very low
False positives (patients incorrectly classified as having acute appendicitis)								50,884 (33,922 to 66,847)	

^a^ Half of included studies were high risk of bias in patient selection; ^b^ I^2^ > 50%; ^c^ Wide range of 95% confidence interval.

## Data Availability

Data was listed in Table 1.

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
