# Peer review of "Can The ‘Speed Bump Sign’ Be a Diagnostic Tool for Acute Appendicitis? Evidence-Based Appraisal by Meta-Analysis and GRADE"

_life, 2022, doi:10.3390/life12020138_

Round 1
Reviewer 1 Report
Authors performed a meta analysis whether speed bump sign would help diagnose acute appendicitis (AA). The results showed that this sign would help rule out AA: however, I have several concerns about this study.
- Authors wrote that the evidence of speed bump sign has not been evaluted. This is true, but the purpose of the study is a little weak, since all four previous articles showed that this method has high sensitivitiy and low specificity.
- Elder patients especially with diabetes are less likely to feel the pain. Does this sign still work for elder patients?
- Was there any difference in sensitivty and specificity between simple and complicated appendicitis?
Author Response
Q1. Authors wrote that the evidence of speed bump sign has not been evaluated. This is true, but the purpose of the study is a little weak, since all four previous articles showed that this method has high sensitivity and low specificity.
Response: We appreciate the reviewer 1’s reminder. We should highlight the aim and advantage of systematic review in the section of introduction. Using systematic review to search and collect all the similar research studies, appraise the risk of bias, and further perform meta-analysis if suitable is the standard strategy in the field of evidence-based medicine. The methodological strength of meta-analysis is more efficient to present a combined result by pooling the sample size to increase statistical power than to report the results of each individual study. We modify the manuscript following reviewer 1’s comment as the red colored words highlighted in the revised edition of manuscript.
Q2. Elder patients especially with diabetes are less likely to feel the pain. Does this sign still work for elder patients?
Response: We thank for the reviewer 2’s notice. The age of patients in the four included studies was relatively young, therefore the results in our study might not be suitably applied to the elderly patients who might had sensory abnormalities. We modify the manuscript following reviewer 1’s reminder in the section of discussion for additional description of the study limitation as the red colored words highlighted in the revised edition of manuscript.
Q3. Was there any difference in sensitivity and specificity between simple and complicated appendicitis?
Response: Due to limited cases, there were no subgroup results between complicated and non-complicated appendicitis in the included studies. Based on the mechanism of peritoneal irritation during the pass of speed bumps, aggravation of abdominal pain might be more strongly triggered in patients with complicated appendicitis. Perhaps the sensitivity might be higher in complicated appendicitis than non-complicated one. Of course, we are not sure the different presentation of severity in speed bump sign between two groups. We consider the specificity for the subgroup of complicated appendicitis might not be changed. To diagnose complicated appendicitis is an annoying issue for physicians, and future studies are warrant to deter-mine the diagnostic value of speed bump sign for patients with complicated appendicitis.

Reviewer 2 Report
The paper in object is well designed in terms of statistical analysis and conclusions do answer the main question. The argument is not original but could be interesting if considering it could help in the diagnosis, reducing the need of further exams( CT scan, US) thus limiting costs. Furthermore could help when while facing in limited resources areas.
Author Response
Comment:
The paper in object is well designed in terms of statistical analysis and conclusions do answer the main question. The argument is not original but could be interesting if considering it could help in the diagnosis, reducing the need of further exams (CT scan, US) thus limiting costs. Furthermore, could help when while facing in limited resources areas.
Response: Thanks for the reviewer 2’s comments. We totally agree the benefit of utility of speed bump sign in limited resources areas. Therefore, we modify the manuscript to highlight this feature with blue colored words in the revised edition of manuscript.

Round 2
Reviewer 1 Report
Authors have responded to my comments. I agree to accept this article.